# Critical Roles of SRC-3 in the Development and Progression of Breast Cancer, Rendering It a Prospective Clinical Target

**DOI:** 10.3390/cancers15215242

**Published:** 2023-10-31

**Authors:** Lokman Varisli, Garrett M. Dancik, Veysel Tolan, Spiros Vlahopoulos

**Affiliations:** 1Department of Molecular Biology and Genetics, Science Faculty, Dicle University, Diyarbakir 21280, Turkey; vtolan@dicle.edu.tr; 2Department of Computer Science, Eastern Connecticut State University, Willimantic, CT 06226, USA; dancikg@easternct.edu; 3First Department of Pediatrics, National and Kapodistrian University of Athens, Thivon & Levadeias 8, Goudi, 11527 Athens, Greece

**Keywords:** breast cancer, SRC-3, estrogen signaling, tumor microenvironment, tumor infiltrating cells

## Abstract

**Simple Summary:**

Breast cancer is one of the most commonly diagnosed malignancies in women and is also a leading cause of cancer related death. Estrogens play crucial roles in the development and progression of breast cancer, and estrogen signaling is generally mediated by estrogen receptors. Steroid receptor co-activator protein family members are transcriptional activators that interact with steroid receptors, including estrogen receptors. SRC-3 is a member of this family and has been shown to be overexpressed and/or amplified in breast cancer. Here, we review and discuss the versatile effects of SRC-3 in breast malignancy and its potential as a therapeutic target.

**Abstract:**

Breast cancer (BCa) is the most frequently diagnosed malignant tumor in women and is also one of the leading causes of cancer-related death. Most breast tumors are hormone-dependent and estrogen signaling plays a critical role in promoting the survival and malignant behaviors of these cells. Estrogen signaling involves ligand-activated cytoplasmic estrogen receptors that translocate to the nucleus with various co-regulators, such as steroid receptor co-activator (SRC) family members, and bind to the promoters of target genes and regulate their expression. SRC-3 is a member of this family that interacts with, and enhances, the transcriptional activity of the ligand activated estrogen receptor. Although SRC-3 has important roles in normal homeostasis and developmental processes, it has been shown to be amplified and overexpressed in breast cancer and to promote malignancy. The malignancy-promoting potential of SRC-3 is diverse and involves both promoting malignant behavior of tumor cells and creating a tumor microenvironment that has an immunosuppressive phenotype. SRC-3 also inhibits the recruitment of tumor-infiltrating lymphocytes with effector function and promotes stemness. Furthermore, SRC-3 is also involved in the development of resistance to hormone therapy and immunotherapy during breast cancer treatment. The versatility of SRC-3 in promoting breast cancer malignancy in this way makes it a good target, and methodical targeting of SRC-3 probably will be important for the success of breast cancer treatment.

## 1. Introduction

Breast cancer (BCa) is one of the most commonly diagnosed malignancies in women and is also a major cause of cancer death [1,2]. Estrogen signaling is a critical mechanism in BCa pathogenesis which involves ligand-activated cytoplasmic estrogen receptors, and the signals transmitted through estrogen receptors (ERs) located in the cell membrane and cytoplasm [3,4,5]. 

The steroid receptor co-activator (SRC) family consists of three transcriptional regulators that enhance the activity of steroid receptors, including ERs [6]. SRC-3 (also known as AIB1, NCoA-3, RAC3, ACTR, TRAM1, and p/CIP) is a member of this family and is involved in tissue homeostasis and development in normal physiology [7,8]. The activity of SRC-3 is generally regulated at the post-translational level and a large number of modification sites have been identified on the primary structure of the SRC-3 protein, including phosphorylation, methylation, acetylation, ubiquitination, and SUMOylation [9,10].

SRC-3 has also been shown to be overexpressed and amplified in BCa and has critical roles in the development and progression of the disease [11,12]. In fact, detailed studies have shown that SRC-3 is involved in BCa pathogenesis in multiple ways by promoting proliferation, migration, invasion, and metastasis in BCa cells [13,14]. SRC-3 is also involved in the creation of an immunosuppressive tumor microenvironment and promotes stemness [15,16]. In regard to cancer treatment, SRC-3 is involved in the development of resistance to both endocrine therapy and immunotherapy in patients with BCa [17].

Given the multiple critical roles of SRC-3 in the development and progression of BCa, the potential of SRC-3 as a possible therapeutic target is high. Therefore, various natural and lab-made artificial molecules have been designed to inhibit SRC-3 activity or reduce its levels in cells, as explained in the following parts. The results of both in vitro and in vivo experiments showed that a decrease in SRC-3 levels or activity both inhibited the malignant behavior of BCa cells and changed the tumor microenvironment to a tumor-suppressing phenotype. Overall, the results of current studies on the inhibition of SRC-3 levels or activity are very strong and promising. Therefore, the development of new therapeutic approaches targeting SRC-3 for the treatment of BCa in the clinic in the near future is a strong possibility. However, in order to achieve this, it will be essential to overcome the current limitations of molecules targeting SRC-3 and to design/develop new nanocarrier systems that will enable the delivery of these molecules to specific cell types such as cancer stem cells (CSCs).

## 2. BCa and Estrogen Action

BCa is the most frequently diagnosed malignant tumor in females and is one of the leading causes of cancer related death in women [1,2]. Although BCa is highly curable in patients with early stages, it is generally incurable in patients with advanced stages who have distant metastases [18]. In fact, BCas are highly heterogeneous tumors with many subtypes, which are generally classified into four groups according to the status of the ER, progesterone receptor (PR), and human epidermal growth factor receptor 2 (HER2) Luminal A, Luminal B, HER2(+), and Triple(−) (TNBC) [19,20]. The Luminal A class of BCa are ER(+) and/or PR(+) and mostly show a low proliferation rate and low Ki-67 levels. The Luminal B class of breast tumors express ER and/or PR, similar to Luminal A tumors, but their proliferation rates are high in concordance with higher Ki-67 levels. HER2(+) tumors are ER and PR deficient but have HER2. In TNBC all three hormone receptors, (ER, PR, and HER2) are deficient. The BCa subclasses show different malignant behaviors, often dependent on the status of hormone receptors, and therefore, the absence or presence of these receptors is also associated with BCa prognosis [21]. For example, while ER(−) tumors are considered to be associated with early recurrence risk, ER(+) tumors are generally associated with late recurrence risk [22,23]. Moreover, although patients with hormone receptor-positive Luminal A and B tumors generally have a favorable prognosis, patients with hormone receptor-negative tumor sub-classes have been shown to have a poor prognosis [24]. Furthermore, there are uncommon histological subtypes of BCa such as invasive micropapillary carcinoma (IMPC) [25]. IMPC tumors are generally considered to be an aggressive subtype of BCa with distinctive histologic and pathologic characteristics and are reported to predominantly exhibit Luminal A and Luminal B characteristics when classified by hormone receptor status, but may also exhibit HER2 and TNBC characteristics in a lower proportion [26,27,28]. It has been shown that IMPC tumors with TNBC subtype characteristics are associated with higher grade tumors, and that estrogen receptor status is positively associated with disease-free survival [25]. Ductal carcinoma in situ (DCIS) is a heterogeneous disease that is recognized as a non-obligate precursor lesion of invasive BCa [29]. The status of ER and PR has also been investigated for the prognostic significance of DCIS, and ER(−) and PR(−) tumors have been associated with increased recurrence [30,31]. Although the status of HER2 has also been investigated in DCIS patients, its clinical and biological significance remains poorly defined [32,33,34]. However, increasing evidence suggests that HER2 overexpression in DCIS may be associated with increased risk of recurrence and worse clinical outcomes [34,35].

Estrogens are steroid hormones that have crucial functions in the development of secondary sexual characteristics in females. However, they are also involved in the pathogenesis of BCa in multiple ways. Estrogens are mainly produced and secreted by the ovaries under the control of the luteinizing hormone (LH)/follicle-stimulating hormone (FSH) axis [36] and act on the target cells through ERs which are called ER-α and ER-β that are encoded by *ESR1* and *ESR2*, respectively [37]. ERs are the members of the nuclear hormone receptor (NHR) superfamily and have diverse functional domains to carry out their functions [38]. In the absence of estrogen, ERs are located in the cytoplasm, inactively bound with various HSP proteins including HSP70 and HSP90, in a complex form [39]. Estrogen activated ER may act in a genomic or in a non-genomic manner in the cells (Figure 1).

In the genomic mode of action, estrogen binding causes conformational changes in the ER that convert it to an active form and dissociate it from HSPs [40]. Consequently, dimerized ER is translocated to the nucleus, generally with some co-regulators including steroid receptor activators (SRCs), and binds the estrogen response elements (EREs) on the regulatory regions of target genes to regulate their expression [41]. Activated ER may also be indirectly involved in the transcriptional regulation of some genes through interactions with transcription factors such as SP1 and Cyclin G2 [42,43,44]. In both mechanisms, the active ER interacts with co-regulators such as SRCs and other molecules to regulate the transcription of target genes [45,46]. In the non-genomic mode of action, membrane located ER, activated through estrogen binding, leads to many changes in the cytoplasm and affects a variety of signaling pathways including NFκB, MAPK, and PI3K/AKT [5]. In addition, GPR30 (also known as GPER1), a G protein-coupled receptor, on the cell membrane has been identified as a membrane estrogen receptor [47,48]. Although the affinity of estrogen to GPR30 isn’t as high as compared to classical intracellular ERs, it triggers a rapid response and then conveys the signal to a number of intracellular signaling cascades through second messenger—dependent or independent mechanisms [49,50].

Although both of the intracellular ERs are expressed in normal breast tissues, the ER-β level is higher, making it the dominant receptor in estrogen/ER signaling [51]. However, in the ER(+) BCa cells the level of ER-β is decreased and ER-α is increased [52,53]. Similarly, an increased ER-α to ER-β ratio, dependent on a decrease in ER-β expression, has also been shown in both uterine myomas and mouse skin tumors and cell lines [54,55]. In this context, although an increased ER-α to ER-β ratio seems to be a general mechanism in both malignant and benign hyperproliferative tissues, it is the major source promoting malignant behaviors in ER(+) BCa cells. Indeed, it has been shown that the changed expression of ER-α is associated with the development and progression of BCa [56]. Many ER-α target genes involved in cell proliferation have been identified, including the genes encoding c-Myc, Cyclin D1, FoxM1, and Greb1 [57,58,59]. Estrogen/ER-α signaling has been shown to also be involved in the BCa cell migration, but this effect is complicated and may act in both inhibitory and activatory roles depending on the co-regulators and downstream mechanisms. Indeed, estrogen suppresses the E-cadherin level through direct binding of ER-α together with several co-repressors to the promoter of *Cdh1* which is the E-cadherin encoding gene [60]. On the other hand, metastasis-associated protein 3 (MTA3), another direct target of ER-α, has been shown to repress the transcription of *Snail 1* which is a well-known repressor of *Cdh1* [61]. Consequently, increased MTA3 levels depending on the estrogen/ER-α axis represses *Snail 1* expression, and therefore, *Cdh1* transcription is restored. Another MTA family member, MTA2, has also been shown to interact with SRC-3, to establish an inhibitory complex, thereby inducing epithelial-mesenchymal transition (EMT) by repressing *Cdh1* expression in ER(+) luminal breast tumors [62]. Therefore, it is plausible to postulate the formation of an inhibitory SRC-3/MTA2 interaction complex under estrogen-absence conditions since SRC-3 expression is negatively regulated by estrogen [63]. Maintaining or restoring of the E-cadherin level on the BCa cells is important in both the maintenance of tissue stability and consequently inhibiting migrative and invasive abilities of cancer cells, and also for the targeting of tumor cells by immune cells, as described in the next sections.

## 3. BCa Tumor Microenvironment 

The tumor microenvironment is a dynamic, complex network composed of cellular and non-cellular components which have crucial roles in the behaviors of tumor cells including invasion, metastasis, and therapy resistance [64]. Indeed, the effects of the tumor microenvironment on both promoting the malignant behaviors and metastasis processes of tumor cells have been known for a long time. Although the tumor microenvironment is highly heterogeneous across cancer types, the cellular content of the microenvironment consists of stromal cells and of tumor-infiltrating cells (TICs), which encompass multiple cell groups including T-cells, leukocytes, monocytes, and tumor associated macrophages (TAMs) [65,66,67]. Tumor-infiltrating lymphocytes (TILs) are important components of TICs which affect tumor cell metabolism directly and determine the nature and phenotype of the microenvironment. However, they are not found uniformly in all BCa subtypes. Although TILs are found at a relatively low rate in the Luminal A and Luminal B types of tumors, they are found to be increased in TNBC and HER2(+) breast tumors [68]. The non-cellular components which surround cellular components are basically composed of growth factors, cytokines, chemokines, extracellular vesicles, and extracellular matrix proteins [65,67]. In fact, the tumor microenvironment as a whole, with both cellular and non-cellular components, is a specialized ecosystem and often acts independently, at least partly, from the rest of the organism. For example, it often has a low oxygen concentration that causes the recruitment of regulatory T-cells (Tregs) to the microenvironment and the inhibition of effector T-cell differentiation, which would otherwise combat cancer cells [69]. In addition, cancer cells secrete fibroblast growth factor (FGF), which causes the recruitment of cancer associated fibroblasts (CAFs) to the tumor microenvironment, where CAFs both change the extracellular matrix and secrete various immunosuppressive cytokines that promote angiogenesis as well as the growth of tumors [64,70].

Although breast tumors generally have an immunosuppressive microenvironment, both the effect of estrogen and the absence/presence of ER have a prominent impact on the composition of infiltrated TILs [71]. In fact, estrogens have multiple effects on the immune system cells including the proliferation, differentiation, and regulation of cytokine production [72,73,74,75,76,77]. For example, estrogen inhibits both the proliferation of CD4+ T-cells and the activities of natural killer (NK) and cytotoxic T lymphocytes (CTLs) to contribute to promoting an immunosuppressive tumor microenvironment [78,79]. However, an immunosuppressive tumor microenvironment is created by the collective actions of many cells, including the tumor cells and some immune system cells. In this context, the secretion of immunosuppressive molecules such as TGF-β and IDO-1 by tumor cells, and the recruitment of immunosuppressive immune system cells, such as Tregs, into the tumor microenvironment are the lead causes in the creation of an immunosuppressive phenotype [80]. Indeed, the amplification and immunosuppressive activities of Tregs were shown to be promoted by estrogen [81,82,83]. The increase in the estrogen-dependent immunosuppressive activities of Tregs is caused by increase in FoxP3 and PD-1 levels [81,82]. Estrogen also induces the production and secretion of TGF-β and IL-10 in Tregs [84,85]. In concordance, the treatment of Tregs with ICI-182780, an ER antagonist, was shown to inhibit both FoxP3 expression and TGF-β production and secretion [86]. 

Tregs are crucial cells in immune homeostasis and act as the suppressors against excess immune response to prevent autoimmune diseases; they are highly increased in the tumors of many types of solid cancers including BCa [87]. Tumor infiltrating Tregs have a different phenotype to normal tissue Tregs and it has been suggested that the interaction of Tregs with the tumor microenvironment probably drives this distinct Treg phenotype [88,89]. Indeed, it has been shown that the immunosuppressive roles of Tregs are associated with both their presence in the tumor microenvironment and with an increase in the inhibitory receptors [90]. Tregs found in the tumor microenvironments are generally associated with poor prognosis since they both suppress effector T-cells and also inhibit the efficacy of chemotherapy and radiotherapy [91]. In concordance, it has been shown that selective depletion of Tregs results in augmentation of the anti-tumor immune response [92].

Tregs may perform their immune suppressor role through several different mechanisms in the tumor microenvironment. Tregs secrete anti-inflammatory cytokines including TGF-β, IL-10, and IL-35 into the microenvironment, and thereby suppress the activity of effector immune cells and consequently anti-tumor immunity [93,94,95,96]. Secretion of these cytokines is important in the creating and sustaining of an immunosuppressive tumor microenvironment. For example, IL-10 secreted by Tregs increases the expression of co-inhibitory receptors, and thereby induces immunosuppression functions of Tregs [96,97]. IL-35 generally acts jointly with IL-10 in the generation and sustaining of the immunosuppressive tumor microenvironment [96]. Although most of TGF-β in the tumor microenvironment is produced by tumor cells and tumor-associated fibroblasts, it can also be produced and secreted by Tregs [98]. TGF-β may affect the tumor microenvironment in multiple ways. For example, TGF-β can induce Treg differentiation from the naive CD4+ T-cells [99]. Tregs, can also directly kill CD8+ T-cells and NK cells by secreting granzyme B and perforin [100,101]. On the other hand, cell to cell interaction is another important mechanism in Treg biology, and Tregs commonly express several co-regulatory receptors to modulate their actions. In this context, ICOS, OX40, and GITR function as co-stimulatory receptors, whereas CTLA-4, Lag-3, Tim-3, TIGIT, PD-1, and KLRG1 function as co-inhibitory receptors [102]. The functional consequences of the interaction between many of these receptors and their ligands (usually found in antigen presenting cells, tumor, or epithelial cells) have been revealed, at least partially.

Another important class of T-cells that are found in the tumor microenvironment is T helper 17 (TH17) cells. Th17 cells are a subset of CD4+ T-cells and their differentiation is mainly regulated by the cytokine milieu; classically, Th17 cells differentiate via TGF-β and IL-6 stimulation [103,104]. In addition, it was shown that IL-1, IL-6, and IL-23 may also induce differentiation of Th17 cells without TGF-β [105]. The signature cytokines of Th17 are IL-17A/IL-17F; RORγt is the main transcription factor in the expression of these cytokines [106,107]. Furthermore, additional transcription factors such as Rorα are also involved in the expression of these cytokines [107].

The primary function of these cells is to regulate host defense responses against extracellular pathogens such as bacteria or fungi [108]. However, Th17 cells have been implicated in the pathogenesis of autoimmune diseases, including multiple sclerosis, rheumatoid arthritis, and inflammatory bowel disease, as the pathogenic factors [109].

Th17 cells also function in the immune responses against the tumors. However, while their roles in the autoimmune diseases are relatively well understood, their functions in cancer pathogenesis are more complicated. In this context, it has been shown that the function and character of Th17 cells are uniquely dependent on the type of cancer, the therapeutic approach used, and the factors that induce their differentiation [110]. Indeed, it has been shown that Th17 cells exhibit a regulatory cell phenotype and show anti-tumor activity in some types of cancer such as liver, prostate cancer, ovarian, and melanoma, while they have a tumor-promoting function in other types such as lung, liver, and pancreatic cancers [111]. IL-17A generally promotes the tumor-promoting activity by affecting the ERK, p38, and NF-κB signaling mechanisms, in an IL-17RA dependent manner [112]. Indeed, it has been shown that tumor-infiltrating Th17 cells promote liver cancer cell migration by increasing the levels of MMP2 and MMP9 in an NF-κB dependent manner [113]. In addition, IL-17 promotes liver cancer cell migration and invasion by increasing the levels of IL-8, MMP2, and VEGF [114]. IL-17 was also shown to promote self-renewal of CSCs in an NF-κB and p38 MAPK dependent manner in ovarian cancer [115].

On the other hand, the antitumor activities of IL-17 have been demonstrated in various types of cancers including colon and gastric cancers. It was shown that inducing Th17 cell differentiation using RORγ agonists increases the levels of IL-17A, IL-17F, IL-22, and GM-CSF, and inhibits PD-L1 levels [116]. In addition, IL-17 promotes an anti-tumor immune response via recruiting T-cells and increasing the activities of NK cells and CTLs [117,118,119]. Similarly, IL-17A has also been shown to recruit immune cells and is considered a marker for a favorable prognosis in esophageal cancers [120].

Th17 cells are not a homogenous cell population and their opposite functions in different cancers rely on their high plasticity [121]. In this context, Th17 cells acquire either an immunomodulatory or a pro-inflammatory phenotype, and their activities are shaped by this. Th17 cells with immunomodulatory phenotype are generally found in peripheral blood, and express IL-17 and immunomodulatory genes associated with Tregs such as CTLA-4 and LRRC32 [122]. They also express various transcription factors such as MAF, AHR, and IKZF3 to regulate and promote IL-10 expression [123]. On the other hand, Th17 cells with a pro-inflammatory phenotype are found at inflammation sites and express IFN-γ in addition to IL-17; they also produce inflammatory cytokines such as GM-CSF, IL-26, CCL20, and IL-22. They also express RORγt and T-bet transcription factors in addition to the receptors IL-12Rβ2 and IL-23R [122,124]. Th17 cells are believed to play a role in the pathogenesis of autoimmune diseases [122]. 

In regard to the roles of Th17 cells in BCa, it was shown that BCa patients with high numbers of tumor-infiltrating Th17 cells have shorter disease-free survival compared to those with lower numbers of tumor-infiltrating Th17 cells [125]. Accordingly, intratumoral IL-17 expression has also been shown to be associated with unfavorable clinical outcomes in BCa [126]. Similar results have been reported in an animal study showing that increased IL-17 expression depended on tumor-infiltrating Th17 cells and associated with advanced tumors; this effect increased expression enhanced angiogenesis and promoted tumor progression [127]. Furthermore, it was reported that IL-17 may be associated with the aggressiveness of BCa and that the levels of circulatory soluble IL-17A and MIF are higher in TNBC, HER2, and Luminal B subtypes compared to the Luminal A subtype, and therefore, an increase in the expressions of IL-17 with MIF may be associated with the progression of BCa [128]. On the other hand, it has also been reported that there is a positive correlation between the Th17 cell population and a favorable prognosis in patients with TNBC, and it has been suggested that the accumulation of Th17 cells may induce an anti-tumor response and thereby prevent BCa progression [129]. Therefore, although Th17 has generally been reported to play a tumor-promoting role in BCa, it appears that this effect may vary depending on the subtype.

## 4. SRC-3 Has Multiple Roles in BCa Pathogenesis 

SRC-3 is a member of the p160 steroid receptor co-activator family and has critical roles in the regulation of normal cell physiology [6]. The SRC-3 protein is mainly regulated functionally at the post-translational level and many modifications have been identified on its primary structure [9,10,130]. Indeed, it has been shown that alterations in functional post-translational modifications of SRC-3 cause systemic changes in growth and metabolism [131]. In particular, phosphorylated SRC-3 interacts with ligand-activated nuclear hormone receptors and then recruits various proteins including histone acetyltransferases through its different domains (Figure 2) [132]. The established complex alters chromatin dynamics and consequently enhances the transcriptional activities of the NHRs [132].

Indeed, the interactions of SRC-3 with protein partners are crucial for it to carry out its functions in cells, and this event is not restricted to the regulation of the transcriptional activities of steroid receptors. There are many reports that identify the SRC-3 interacting proteins in concordance with their multifunctional nature. For example, SRC-3 interacts with various proteins involved in the regulation of posttranslational modifications of histones, including CBP/p300, p/CAF, GCN5, CARM1, and PRMT1, to facilitate chromatin remodeling, as summarized above [6,133]. In addition, SRC-3 has been shown to interact with numerous transcription factors in accordance with its co-activator function [134]. Although only a few of these interactions, particularly in the breast cancer context, are briefly discussed below with their functional consequences, it is worth noting that the number of interaction partners of SRC-3 in the transcription factor context is much greater. An interaction between SRC-3 and HIF-1α potentiates the transcription of *Mif,* and thereby promotes survival in BCa [135]. Furthermore, an interaction between SRC-3 and E2F1 increases the transcription factor activity of E2F1 and promotes proliferation in ER(−) BCa cells [136]. SRC-3 interacts with ETS family members including PEA3, ER81, ETS1, and ETS2. The interaction of SRC-3 with PE3 on the MMP2 and MMP9 promoters increases their expressions and thereby promotes the invasion capacity of BCa cells [137]. An interaction between SRC-3 and ER81 on the MMP1 promoter drive the expression of MMP1 and this event might be associated with the metastatic ability of BCa cells [138]. SRC-3 interacts with ETS-1 and ETS-2; growth factors increase the interaction between SRC-3 and ETS2 and this event results in an increase in the HER2 levels in BCa cells [139].

SRC-3 also interacts with critical signaling nodes such as AKT and ERK. It was shown that SRC-3 promotes migration and invasion via interacting with AKT in trophoblast cells [140]. ERK3 interacts with SRC-3, which it phosphorylates, and thereby promotes lung cancer cell invasion [141].

SRC-3 is essentially involved in the regulation of metabolic homeostasis similar to other SRC family members. SRC-3-deleted or overexpressed mouse models have been developed by several research groups to elucidate the roles of SRC-3 in normal cellular physiology and crucial data have been derived. For example, SRC-3 is involved in the regulation of adipocyte differentiation, and white adipose differentiation is impaired in SRC-3 deleted mice [142]. SRC-3 is also involved in the regulation of energy metabolism, in preventing obesity, and increasing glucose tolerance, by augmentation of mitochondrial activity [143]. Furthermore, the observations of SRC-3-deleted mice show a pleiotropic phenotype including abnormalities in the mammary gland development and a reduction in reproductive function [8].

### 4.1. SRC-3 Affects the Tumor Microenvironment

SRC-3 may affect the tumor microenvironment in multiple ways by affecting many immune system cells, as well as tumor cells. Although these effects generally result in an immunosuppressive phenotype, they may also contribute to the creation of an inflammatory environment. Tregs have a special place in the SRC-3-induced immunosuppressive tumor microenvironment. Indeed, Tregs have a strong SRC-3 expression and a high SRC-3 level is required for their immunosuppressive functions as described above [15]. In this context, it can be speculated that SRC-3 has a crucial role in establishing an immunosuppressive microenvironment in BCa. Although the factors that induce SRC-3 expression in Tregs are not yet understood, it is possible that this regulation may be dependent on some common mechanisms including the estrogen, retinoic acid (RA), and TGF-β signaling mechanisms. Silencing or inhibiting of SRC-3 in Tregs causes a decrease in the expression levels of both FoxP3 and PD-1 encoding genes [15]. FoxP3 and PD-1 levels are directly related to the immunosuppressive abilities of Tregs, and estrogen up-regulates the expressions of the genes that encode these proteins [81,82]. However, estrogen has the opposite effect on SRC-3 transcription and activity, and although estrogen inhibits SRC-3 transcription, it induces SRC-3 phosphortylations and consequently activates the SRC-3 protein [10,63]. Consequently, the phosphorylated SRC-3 binds to ERs to potentiate its genomic or non-genomic functions and thereby contributes to estrogen action. RA and TGF-β have important roles in the generation and differentiation of Tregs; RA induces FoxP3 expression in a TGF-β dependent manner and consequently promotes generation of Tregs [144,145]. Indeed, it has been shown that RA is a crucial factor in the TGF-β-mediated immune response that inhibits the IL-6-mediated induction of Th17 and promotes Treg differentiation [146]. Furthermore, it has been shown that RA and TGF-β increases SRC-3 transcription [10]. Therefore, it seems reasonable to hypothesize that SRC-3 may play a role in the RA- and TGF-β-mediated generation and/or differentiation of Tregs and Th17 cells. Indeed, SRC-3 has been shown to play an essential role in Th17 biology [147,148]. It has been demonstrated that SRC-3 interacts with RORα and RORγt and is involved in the activation of the expressions of RORγt-associated Th17 genes via IL-1/ILR1 signaling; thereby regulating pathogenic inflammation [149,150]. In concordance, Wang et al. recently showed that SRC-3 could shape the multiple myeloma microenvironment by inducing IL-17 expression in γδ T-cells. Mechanistically, they showed that the hypoxic microenvironment conditions in the multiple myeloma bone marrow niche stimulate SRC-3 expression in γδ T cells, and consequently SRC-3 interacts with RORγt and promotes IL-17 transcription. In concordance, they also demonstrated that inhibition of SRC-3 activity suppresses IL-17A expression in γδ T cells, reduces the multiple myeloma progression in mouse models and enhances the efficacy of bortezomib [151]. In further concordance, an association has been reported between high SRC-3 levels and poor outcomes in multiple myeloma patients treated with bortezomib, which suggests that targeting SRC-3 may be a promising approach to help overcome drug resistance [152]. Mechanistically, this effect was regulated mainly through NSD2 binding to SRC-3 to stabilize it [152]. NFκB has been shown to bind to the SRC-3 promoter and up-regulate its expression in response to TNF-α [153]. Although NFκB signaling is generally considered to be the mechanism that induces the differentiation of effector T-cells, it also promotes FoxP3 expression and has a role in the generation of Tregs [154,155,156]. SRC-3 expression has been shown to decrease in an AKT/mTOR-dependent manner in hypoxia conditions in preeclampsia, a complication of pregnancy [140]. Although it is unknown whether AKT/mTOR-dependent regulation of SRC-3 expression is a general mechanism in cells, including Tregs, it is possible that it is a general mechanism in the regulation of SRC-3 expression. 

Furthermore, the effects of SRC-3 in the induction of an anti-inflammatory environment have also been shown. *Chen* et al. have shown that SRC-3 inhibits the inflammation, and deletion of SRC-3 in mice which results in increased production of inflammatory cytokines such as TNF-α, IL-1β, and IL-6, and consequently, increased inflammation in the colon [157]. In concordance, induction of SRC-3 activity through the small molecule MCB-613 results in the enrichment of anti-inflammatory macrophages in mice [158]. The anti-inflammatory effect of SRC-3 has also been observed in vitro. In addition, stimulation of SRC-3 through MCB-613 in the RAW 264.7 macrophages results in decreasing expression of pro-inflammatory cytokine mRNAs including TNF-α, IL-1β, and IL-6 [158]. In concordance, it has been shown that LPS treatment leads to the increased secretion of pro-inflammatory cytokines, such as TNF-α, IL-6, and IL-1β, in SRC-3 deleted macrophages compared to wild-type macrophages [159]. Interestingly, the transcription of pro-inflammatory cytokines is nearly unchanged in SRC-3 deleted macrophages compared to wild type, but the translational efficiency of these cytokine mRNAs has been increased [159]. It has been shown that SRC-3-dependent regulation of this effect occurs at the post-transcriptional level, and SRC-3 exerts this effect by promoting the binding of some translational repressors to the 3’ UTR region of TNF-α mRNA to inhibit its translation. Furthermore, macrophages from SRC-3-deleted mice produce a high level of TNF-α protein in response to LPS stimulation without changing the TNF-α mRNA level [160].

SRC-3 may also affect the phagocytosis abilities of macrophages. It has been shown that the levels of the scavenger receptor A and catalase are lower in SRC-3 deficient macrophages, compared to wild-type macrophages [160]. In this context, SRC-3 directly contributes to the regulation of catalase transcription, and SRC-3 deficiency results in a decrease in catalase expression [160]. Catalase is an important enzyme in the regulation of reactive oxygen species and is responsible for the conversion of H2O2 to H2O [161]. It has been shown that both the ROS level and apoptotic index are higher in SRC-3 deleted macrophages compared to wild types [160]. Indeed, other studies also confirmed an inhibitory role of SRC-3 in both intrinsic and extrinsic apoptotic pathways [162,163]. 

On the other hand, it seems that SRC-3 is involved in both the activation and recruitment of neutrophils through the regulation of CXCL-2, in a NFκB dependent manner, and thereby SRC-3 may contribute to the creation and regulation of an inflammatory environment, at least in the neutrophil context [164]. Consistently, SRC-3 was shown to be an NFκB co-regulator that promotes NFκB-mediated transcriptional activity, and this activity is regulated by phosphorylation by IκB kinase [165,166]. The role of SRC-3 in regulation of NFκB was further supported by demonstration of a direct interaction between SRC-3 and Rel-A [166]. NFκB signaling is known to inhibit apoptosis, and therefore, SRC-3 dependent inhibition of apoptosis may be related to the activation of NFκB, at least partly. Moreover, SRC-3 is not only a co-activator for NFκB but is also a direct target, and inflammatory cytokines induce SRC-3 expression via direct binding of NFκB to the SRC-3 promoter [153]. It is probable that a feedback loop operates between SRC-3 and NFκB because SRC-3 also represses the translational efficiency of pro-inflammatory cytokines including TNF-α, IL-6, and IL-1β, and this effect is abolished in SRC-3 deficient mice, as described above [159].

### 4.2. SRC-3 Promotes Stemness

SRC-3 was shown to drive the CSC phenotype, in concordance with its EMT promoting roles (Figure 3) [16]. In this context, cytoplasmic PELP1/SRC-3 complexes were shown to mediate the expansion of breast CSCs [167]. Indeed, PELP1/SRC-3 complexes regulate CSCs through modulating metabolic adaptation-associated gene expression programs [168]. Furthermore, SRC-3 is required for the maintenance and induction of the CSC phenotype; consequently, treatment of BCa cells with an SRC-3 inhibitor decreases SRC-3-induced CSCs in BCa [16,169]. The SRC-3 level is positively associated with ALDH+ CSCs in BCa [16]. ALDH1+ CSCs are especially important since they are associated with both tamoxifen resistance and early recurrence after anti-estrogen therapy in breast tumors [170,171]. Although use of disulfiram, an ALDH inhibitor, results in activating T cell immunity and consequently in the clearance of breast CSCs, it is not yet known whether this effect is associated with effects on SRC-3 [172,173]. Furthermore, SRC-3 interacts with SOX-2 and promotes its transcriptional activity [174]. SOX-2 expression increases during the development of tamoxifen resistance and a high SOX-2 level is important to maintain CSCs in BCa [175,176]. SRC-3 interacts with estrogen-related receptor β (ESRRB) and functions as a co-activator in inducting and sustaining embryonic stem cell (ESC) renewal and pluripotency [177,178,179]. Indeed, SRC-3 was shown to induce the expression of self-renewal and pluripotency related genes, including KLF-4, in ESCs [180]. Furthermore, SRC-3 is involved in the regulation of Hematopoietic stem cells (HSCs) by modulating their mitochondrial metabolism [181].

### 4.3. SRC-3 Promotes Malignant Behaviors of Tumor Cells

The implication of SRC-3 in the development and progression of many types of cancers has been reported [14,182]. Indeed, many reports have shown that SRC-3 is involved in carcinogenic processes via multiple pathways (Figure 3). However, SRC-3 has been most extensively studied in BCa. The story of a relationship between SRC-3 and BCa started about 20 years ago and SRC-3 is considered a proto-oncogene since its overexpression leads to BCa in mice [183]. Although both overexpression and amplification of SRC-3 are reported in BCa, its overexpression is much more common compared to gene amplification [11,184,185]. Furthermore, it has been shown that SRC-3 levels are higher in the advanced stages of the disease and that higher SRC-3 levels are associated with poor prognosis in ER(+) BCa [11,186,187,188,189,190]. The effects of SRC-3 in BCa pathogenesis were shown in mice in which SRC-3 was deleted or overexpressed. It was demonstrated that elevating SRC-3 abundance results in hyperplasia and consequently breast adenocarcinoma in mice [183,191,192]. Interestingly, even moderate overexpression of SRC-3 causes pre-malignant transformation in the mammary epithelium [193]. Conversely, SRC-3 deficiency inhibits both v-Ha-ras and chemical carcinogen-induced BCa [194,195]. Furthermore, SRC-3 directly interacts with ER-α in the presence of estrogen, recruits other co-regulators, and consequently increases the transcriptional activity of ER-α to promote cell proliferation [196,197,198]. Thereby, SRC-3 is involved in the pathogenesis of ER(+) BCa and promotes the malignant behavior of overexpressing cells. In this model, SRC-3 is the primary co-regulator for ER-α activity, and its binding allows the sequential binding of secondary co-regulators which are p300/CBP and CARM1 [199]. However, SRC-3 may interact with the mutant estrogen receptor, which is activated in a ligand-independent manner (in the absence of estrogen) [200,201]. In addition, if we discuss IMPC tumors in terms of SRC-3 activity and levels, although the direct effects of SRC-3 in IMPC pathogenesis are not yet known, it is highly likely that it is tumor-promoting, since it both controls the expression of HER2 and is a co-activator of the ER. HER2 status/level is a well-known prognostic biomarker for invasive BCa and its level was shown to be increased in SRC-3-overexpressing BCa cells [189,202,203]. Similarly, the SRC-3 level is higher in DCIS lesions compared to the corresponding normal breast tissue, and an elevated SRC-3 level in DCIS lesions causes an increase in the HER2 and HER3 levels and augments their corresponding signaling activities [204]. Furthermore, SRC-3 overexpression promotes ER(+) ADH lesions which have been considered as the earliest DCIS-related lesions in vivo [205]. In concordance, conditional knock out of SRC-3 results in a significant reduction in the populations of breast cancer initiating cells and myoepithelial progenitor cells, and consequently a decrease in the DCIS lesions [204].

It has been shown that SRC-3 is also involved in the production and secretion of growth factors, and thereby is involved in the regulation of growth factor signaling. For example, SRC-3 overexpression results in an increase in the IGF-I mRNA and protein levels, as well as the components of the IGF-I signaling mechanism, such as IGF-I receptor β (IGF-IRβ) [8,183,206]. In addition, the SRC-3 expression level is positively correlated with HER2, and this event is associated with tamoxifen resistance [189,203]. In concordance, breast tumorigenesis induced by HER2 was completely inhibited in SRC-3 deficient mice [202]. SRC-3 was implicated in the migration, invasion, and metastasis processes: it was namely shown that SRC-3 overexpression results in an increase in the MMP-7 and MMP-10 levels and thereby promotes metastasis [137]. SRC-3 promotes FAK activation, and also functions as an adapter molecule between EGFR and FAK and consequently promotes cell migration in BCa [207,208]. SRC-3 also promotes EMT in cancer cells through the classical cadherin switching mechanism, by which E-cadherin is replaced by N-cadherin [16]. This transition mechanism is crucial for tumor cells to gain migrative abilities and is considered as one of the initial steps in the invasion and metastasis processes of cancer cells [209]. 

Rohira et al. have shown that SRC-3 overexpression induces *Snail 1* and *Snail 2* expressions and thereby decreases E-cadherin level, and in concordance, Vimentin and N-cadherin levels increase in SRC-3 overexpressing cancer cells [16]. E-cadherin is a glycoprotein in epithelial cells and is crucial for the establishment of adherens junctions between neighboring cells [210]. Each E-cadherin molecule has a large extracellular region, a transmembrane region, and a short cytoplasmic domain [211]. The extracellular region consists of five extracellular cadherin domains and interacts with the extracellular region of cadherin in neighboring cells. The cytoplasmic domain of E-cadherin interacts with the cytoskeleton through catenin proteins. E-cadherin loss is observed in the advanced stages of many cancers, including BCa, and this event is a strong marker of EMT [212]. Cadherin switching in advanced stages of cancers is generally associated with an increase in the invasive and metastatic potential of cancer cells, and this event may be the result of various mechanisms that are triggered by genetic or epigenetic alterations [212,213]. Furthermore, it may also be a result of therapeutic approaches, such as androgen deprivation therapy in prostate cancer [214].

## 5. SRC-3 May Contribute to Therapy Resistance in Multiple Ways in BCa

### 5.1. SRC-3 May Contribute to Hormone Therapy Resistance

Considering that more than 70% of BCa patients have ER(+) tumors, and ER signaling plays a crucial role in the development and progression of BCa, clinically targeting this mechanism for the treatment of BCa is quite reasonable, and indeed this therapeutic approach has proven successful in most patients [215]. However, although the patients are mostly successfully treated with these approaches, resistance often emerges after long term exposure [216,217]. One of the mechanisms of acquired resistance is the acquisition of mutations of ER-LBD which may cause ligand-independent activation of the receptor [218,219]. A role of SRC-3 in ligand-independent activation of the ER has been demonstrated: it was shown that SRC-3 interacts more potently with the ER-LBD^mut^ compared to ER-LBD^wild-type^ under hormone-deprived culture conditions created to mimic estrogen-deprivation therapy [220]. SRC-3 may also be involved in endocrine resistance via interacting with estrogen-related receptor α (ERR-α), and ERR-α/SRC-3 complexes may control the expression of estrogen regulated genes in a hormone independent manner [221].

The current therapies that target ER signaling for the treatment of BCa are selective ER modulators (SERMs), selective ER downregulators (SERDs) and aromatase inhibitors (AIs) [222,223]. SERMs are anti-estrogen molecules that compete with estrogen in binding to ER-α and thereby inhibit ER-α-dependent signaling mechanisms [224]. SERDs are antagonists of ERs and their affinities to ER are stronger compared to SERMs. SERD agents generally inhibit the transcriptional activity of ER-α more strongly, compared to SERM agents, and their inhibition mechanisms include promoting proteosomal degradation of ER-α, and disrupting its dimerization/nuclear translocation [225,226,227]. The aromatase enzyme catalyzes the last step of the mechanism of conversion of androgen to estrogen, and therefore, AIs are used to inhibit estrogen induced cell proliferation through blocking this biochemical pathway in postmenopausal women with ER(+) BCa [228]. 

Tamoxifen is the best known SERM agent used in the treatment of ER(+) BCa, and the effects of SRC-3 on tamoxifen in the treatment of BCa have been relatively well studied compared to all other SERM, SERD, and AI group molecules. It was shown that elevated SRC-3, as observed in most BCa patients, inhibits tamoxifen activity, and thereby renders anti-estrogen treatment inefficient [189]. Interestingly, SERM group molecules, including tamoxifen, 4-Hydroxytamoxifen, and raloxifene, increase SRC-3 stability [63,229]. Although tamoxifen increases the SRC-3 level indirectly through induction of TGF-β activity, increased SRC-3 further interacts with ER-α and this event is linked to tamoxifen resistance in ER(+) BCa [63,189]. In concordance, the silencing of SRC-3 results in re-sensitization of ER(+) BCa cells to tamoxifen, and thereby in treatment success [230]. The role of SRC-3 in this mechanism has been shown mechanistically. It was specifically shown that tamoxifen/ER complexes directly bind to the promoter of *Erbb2*, which is the HER2 encoding gene. However, PAX-2 must also be present in the complex to repress, and thereby limit, HER2 expression [231]. Nevertheless, SRC-3 competes with PAX-2 for participation in these complexes and if the SRC-3 level is high, as observed in BCa, the complex includes SRC-3 instead of PAX-2, causing an increase in HER2 levels and consequently tamoxifen resistance [231]. Indeed, increased HER2 expression and activity has been associated with hormone therapy resistance, further metastatic potential, and overall poor prognosis in BCa [232]. In this context, a positive correlation has been observed between the SRC-3 mRNA level and HER2 status/activity in BCa [233]. Furthermore, HER2 may also be involved in the regulation of co-activator function of SRC-3 by phosphorylating it, and thereby increasing the activity of SRC-3 [234]. Indeed, it was shown that tamoxifen treatment results in binding of SRC-3 to the promoter of the PS2 gene, which is a direct target of ER, in HER2-overexpressing, tamoxifen-resistant cells [235]. Finally, a model was proposed to explain tamoxifen resistance in ER(+) BCa cells, based on the increased levels of SRC-3 and HER2, dependent on tamoxifen treatment. In the proposed model, tamoxifen acts as an agonist on ER in the ER(+) BCa cells that have high SRC-3 and HER2 expressions, and therefore these cells develop de novo tamoxifen resistance [189,235]. Elevated SRC-3 is also associated with herceptin resistance in HER2 overexpressing BCa cells. Lahusen et al. have shown that SRC-3 regulates EGFR phosphorylation on multiple sites including autophosphorylation sites, and the silencing of SRC-3 results in a decrease in the total tyrosine phosphorylation on the EGFR, and also in the EGF induced HER2 activation [236].

### 5.2. SRC-3 May Contribute to Immunotherapy Resistance

SRC-3 has immunomodulatory activities that contribute to establishing a tumor-promoting immunosuppressive microenvironment. Indeed, it was shown that SRC-3 may contribute to immunotherapy resistance through the regulation of the immunosuppressive functions of Tregs which are pivotal cells in the creation of an immunosuppressive tumor microenvironment, as described above. SRC-3 expression is high in Tregs, and it has critical roles in regulating the gene expression of these cells [15]. Inhibition of SRC-3 in breast cancer was shown to weaken the immunosuppressive functions of Tregs, consequently leading to the establishment of a tumor-suppressive microenvironment [15]. In concordance, permanent eradication of an aggressive breast cancer model was demonstrated in Treg-cell-specific-SRC-3-deleted mice [237]. Furthermore, deletion of SRC-3 in immune-intact mice or inhibition through a chemical inhibitor results in an anti-tumor microenvironment, and consequently suppresses BCa progression [238]. On the other hand, CXCL-9, Mip-1α, and IFN-γ levels significantly increase in SRC-3-deficient mice [237]. CXCL-9 is an IFN-γ inducible chemokine that attracts various CXCR-3-expressing effector immune cells including CD8+ and CD4+ T-cells and also NK cells, and thereby changes the tumor microenvironment to an anti-tumor phenotype [239,240,241]. Indeed, CXCL-9 overexpression leads to the recruitment of T-cells as well as the inhibition of tumor growth and metastasis in an animal experiment [242]. In concordance, high CXCL-9 levels correlated with an increase in the infiltrating anti-tumor immune cells and also with a better response to chemotherapy in BCa patients [243,244]. 

SRC-3 decreases E-cadherin expression and increases N-cadherin expression, as discussed above [16]. Although this event has been discussed for the epithelial cancer cells in the previous section, it is also important for the success of TIL related immunotherapy. Indeed, the current literature suggests that the presence or absence of E-cadherin on tumor cells may be important in the regulation of immunomodulatory mechanisms, at least in terms of the anti-tumor immune response activities of TILs. This mechanism is based on the fact that E-cadherin is the interaction partner of CD103 which is expressed in some immune system cells, such as NK and effector T-cells, and the interaction between CD103 and E-cadherin activates the cytotoxic functions of these cells (Figure 4). 

CD103 is a heterodimeric transmembrane protein expressed by several immune system cells including CTLs, tissue resident T lymphocytes (TRMs), and Tregs [245,246]. Tregs have an immunosuppressive function as discussed above, whereas CTLs attack the tumor cells and perform crucial functions in the anti-tumor response. TRMs are a special population of CTLs and are involved in the protection of epithelial tissues against viruses [247]. CD103 interacts with E-cadherin expressed in epithelial cells and has roles in the retention of immune system cells within epithelial tissues [248,249]. Indeed, CD103 is differentially expressed in TILs, and the targeting CD103 or E-cadherin by antibodies or genetic approaches inhibits TCR-mediated killing of tumor cells [250]. A heterophilic interaction was shown between the MIDAS motif of CD103 in domain I and the EC-1 domain of E-cadherin [251]. Therefore, in the case of a loss of E-cadherin in tumor cells as in advanced stages of cancers, the tumor-infiltrating immune cells are also decreased, as expected [252,253]. CD103 expression is induced by TGF-β and this induction is stronger in CD8+ T-cells such as CTL and TRMs compared to CD4+ Tregs [254,255]. It has been shown that TGF-β regulates CD103 expression at the transcriptional level. Mechanistically, TGF-β, which is abundantly present in the tumor microenvironment, activates the classical TGF-β/Smad pathway in cells that express CD103, and consequently, transcription of the CD103 encoding gene, *itgae*, increases [256]. In this way, increased CD103 abundance on infiltrating T-cells results in stronger binding of these cells to E-cadherin found on tumor cells [257]. However, TGF-β, produced by tumor cells acts as an immunosuppressive factor that helps cancer cells escape from the immune response by inhibiting the expression of molecules involved in the CTL-mediated tumor cytotoxicity such as perforin, granzyme A, granzyme B, Fas ligand, and IFN-γ [258]. Nevertheless, it is also involved in the migration of T-cells towards epithelial tumors and in promoting the anti-tumor activities of tumor-infiltrating CD8+ T-cells [259,260]. Moreover, TGF-β increases both CD103 expression and its affinity to interact with E-cadherin in an ILK phosphorylation-dependent manner [259].

Another receptor that binds to E-cadherin is killer cell lectin-like receptor G1 (KLRG1), a membrane-spanning glycoprotein expressed on some subsets of NK and T cells [261]. KLRG1 has an extracellular C-type lectin-like domain that can interact with N-cadherin and R-cadherin in addition to E-cadherin [262,263]. KLRG1 is an MHC-independent inhibitory receptor that, when interacting with cadherin molecules on target cells, inhibits TCR signaling and consequently the effector functions of NK and CD8+ T-cells [262,263,264,265,266,267]. Although CD103 and KLRG1 share the same ligand, they have opposite effects on effector T cells. The expression of KLRG1 is also controlled by TGF-βs, like CD103, but this regulation results in the repression of KLRG1 expression, in contrast to CD103 [268]. Therefore, it can be speculated that KLRG1(+) cells should be underrepresented in the tumor microenvironment due to the high TGF-β concentration in the milieu. Indeed, the proportion of CD8+ TILs expressing KLRG1 was shown to be significantly lower in melanoma and renal cell carcinoma [269,270]. 

Furthermore, a negative association has been shown between N-cadherin level and the success of TIL-related tumor immunotherapy. It was demonstrated that N-cadherin increases PD-L1 and IDO-1 levels in an IFN-γ-R1/Jack/Stat signaling-dependent manner in TILs [271]. PD-L1 and IDO-1 induce apoptosis in T-cells, and therefore it was suggested that their inhibition may be useful to increase the success of TIL-related tumor immunotherapy [272,273]. Indeed, Sun et al. have shown that N-cadherin deficiency converts the tumor microenvironment from immunotherapy resistant to responsive through decreasing PD-L1 and IDO-1 levels, and by inhibition of effector Treg production [271]. Similar results were reported by Kolijn et al., namely that EMT causes an increase in the Treg numbers, and also increased IDO-1 levels [274]. In concordance, gene expression analyses from various cancer datasets have shown that the EMT signature is positively associated with immunosuppression signatures, but is negatively correlated with the signature of CD8+ TILs [275]. Taken together, these data suggest that high SRC-3 causes EMT, leading to the recruitment of Tregs into the tumor microenvironment and to an increase in IDO-1 expression, thereby contributing to the development of an immunosuppressive microenvironment and thereby to failure of immunotherapy in BCa.

## 6. SRC-3 Is a Promising Target to Overcome Therapy Resistance in BCa

All the data summarized here suggest that SRC-3 is a proto-oncogene that is involved in BCa pathogenesis via multiple pathways. Therefore, SRC-3 has been suggested as a promising target to overcome therapy resistance in BCa [276,277]. A number of established markers modify the association of SRC-3 with disease outcome for breast cancer: HER2 status, menopause status, ER-status, estrogen-dependence, and triple negative status. Overexpression of SRC-3 in HER2-positive breast cancer is associated with resistance to tamoxifen therapy and decreased disease-free survival [278]. Post-menopause status, however, modifies the risk conveyed by SRC-3 overexpression, converting it to a positive marker, a predictive marker of tamoxifen benefit [279]. Initially SRC-3 was shown to have a role in the estrogen-dependent proliferation of breast epithelial cells; this was because SRC-3 was implicated in the regulation of estrogen-dependent effects on breast cancer development and progression [17]. However, SRC-3 also affects the growth of hormone-independent breast cancer and SRC-3 levels are limiting for IGF-1-, EGF-, and heregulin-stimulated biological responses in breast cancer cells, and consequently, the PI3 K/Akt/mTOR and other EGFR/HER2 signaling pathways are controlled by changes in SRC-3 protein levels [280].

In the meantime, an isoform of SRC3, namely AIB1Δ4 was implicated in the progression of early-stage TNBC: a minor subset of early-stage breast cancer cells expressing AIB1Δ4 enables bulk tumor cells to become invasive, suggesting that selective eradication of this population could impair breast cancer metastasis. Cellular cross-talk was inhibited by the PPARγ agonist efatutazone, but was enhanced via treatment with the GR agonist dexamethasone [281].

The cellular level of the SRC-3 protein is mainly regulated by both ubiquitin-dependent and -independent proteasome degradation mechanisms; however it can be also degraded by non-proteasome dependent mechanisms [229,282]. In this context, the phosphorylation of SRC-3 on S505 and S509 residues by GSK3, and then ubiquitination with SCFFbw7α, was shown [283]. Moreover, SRC-3 is phosphorylated by cell treatment with RA on S860, then ubiquitinated with the CUL-3-based E3 ligase, and consequently degraded [284]. In this regard, it was suggested that molecules such as Gambogic acid and Thevebioside that promote SRC-3 degradation can be used to increase treatment success in cancers, in addition to standard therapy [285,286]. Verrucarin A is another SRC-3 degradation promoting molecule [287]. Verrucarin A probably controls the upstream mechanisms that promote SRC-3 degradation, as discussed above for other molecules, since there isn’t a direct interaction between Verrucarin A and SRC-3 [287]. Although Gossypol has been identified to directly bind to SRC-3 and to lead to its degradation in a proteasome independent manner, it seems that it isn’t a specific inhibitor for SRC-3 [288]. The cardiac glycoside Bufalin is another inhibitor that directly binds to SRC-3 and promotes its degradation. It has been shown that treatment with Bufalin results in the degradation of SRC-3, and also inhibits growth of cancer cells at very low concentrations [289]. However, Bufalin is non-specific for SRC-3, like Gossypol. All the present literature demonstrates that decreasing SRC-3 protein levels using degradation mechanisms may be beneficial in cancer treatment. However, the molecules used for this purpose generally target the entire degradation network, as described above, and are not specific to SRC-3 degradation. Therefore, an approach that can specifically promote the degradation of the SRC-3 protein would be of great benefit. PROTAC is a novel, small-molecule technology to induce ubiquitination and degradation of target proteins [290]. Various degraders have been designed and successfully used to specifically degrade many proteins, including PD-L1, using PROTAC technology [291,292,293]. On this point, the development of degraders that specifically target SRC-3 and promote its proteasomal degradation will be a promising approach.

SRC-3 inhibitor-2 (SI-2) is an non-natural molecule that has been developed in the Lab as an effort of a multidisciplinary study, and it has been shown that SI-2 selectively inhibits SRC-3 expression at both the mRNA and protein levels [277]. Although SI-2 has a short half-life, its low nanomolar activity has made it a promising candidate [14,277]. Song et al. have shown that SI-2 significantly repressed BCa cell proliferation in vitro, and inhibited breast tumor growth in a xenograft model [277]. Furthermore, inhibition of SRC-3 by SI-2 also inhibits the immunosuppressive functions of Tregs and their tumor infiltrations, but causes an increase in the CTL and NK cells, and consequently changes the tumor microenvironment from immunosuppressive to tumor-suppressive [15,237,238]. SI-2 treatment also targets TIC populations and blocks EMT [16]. The same research group has developed the molecules SI-10 and SI-12 with longer half-lives on the basis of the SI-2 scaffold [294]. SI-10 and SI-12 inhibit the malignant behavior of BCa cells and the growth of breast tumors in xenograft models, and suppress the growth of BCa in PDX organoids [294]. It seems that SI-2-based molecules are promising agents for specifically inhibiting SRC-3 in BCa, and we will probably discuss them further in the next few years. On the other hand, it was shown that salinomycin directly inhibits SRC-3 transcription and increases the sensitivity of BCa cells to tamoxifen [295]. Although salinomycin inhibits the malignant behaviors of BCa cells and kills CSCs, it isn’t a specific SRC-3 inhibitor [296]. In this regard, some well-designed nanocarriers can be used to effectively deliver SRC-3 inhibitors into rapidly proliferating cells, such as CSCs. Recent studies have shown that KU-55933, an ATM inhibitor [297], and chloroquine, a lysosome inhibitor [298], can be effectively delivered to breast CSCs using the triphenylphosphonium-functionalized hyperbranched polyethylenimine nanoparticles (PTPP) [299,300]. Similarly, PTPP or PTPP-based nanocarriers could be promising approaches to effectively deliver SRC-3 inhibitors to cells.

There are also nucleic acid-based approaches used to target SRC-3. For example, it has been shown that AY-3, a DNA aptamer, interacts with SRC-3 and abrogates its interaction with p300 [301]. The SRC-3/p300 association is important for the transcriptional activity of ER-α, and the disruption of this interaction inhibits the transcriptional activity of ER-α, and thereby ER-α-promoted malignant behaviors in BCa cells [199]. Therefore, further investigation for the new aptamers that would disrupt and/or abrogate SRC-3 activity and designing new nanocarrier systems that will effectively deliver these aptamers into CSCs and tumor cells will be important to inhibit/overcome SRC-3-promoted malignant behaviors in BCa.

SRC-3 has pleiotropic effects in normal physiology, including cell proliferation, survival and metabolism, along with vasoprotection, female reproductive function, and puberty [17]. SRC-3 null mice displayed delayed puberty, slowed mammary gland growth, reproductive malfunction, and dwarfism due to alterations in the IGF-1 (insulin-like growth factor 1) signaling pathway [7,8]. SRC-3 systemic inhibition would therefore be expected to have a number of effects that depend on the age of the organism. In a growing organism systemic inhibition would lead to growth defects and to deficient reproductive capacity, while in a mature organism systemic inhibition of SRC-3 would be expected to have metabolic effects and lead to endocrine malfunction by altering the balance of steroid receptor signaling pathways; these effects may extend to the central nervous system, where SRC-3 affects amino acid levels [7,302,303].

The Bert O’ Malley group has, in fact, developed an SRC-activator that experiments show promotes wound healing/tissue regeneration in diverse tissues that include heart and brain [158,304]. In fact, part of this effect can be attributed to SRC-3, which was shown to promote the function of regulatory T-cells that are anti-inflammatory and protect regenerating tissue [15]. This very function of SRC-3 is also a key to its promotion of cancer progression via the inhibition of the anti-tumor immune response [237].

The answer, therefore to targeting the multifaceted functions of SRC-3 in the human organism, is on the one hand a space and time restricted intervention that is focused on the site of the malignant tumor, and on the other hand an intervention that ideally lasts only as long as it is needed for the organism to regain control of the tumor by the restoration of the antitumor immune response.

## 7. Conclusions and Future Directions

BCa is one of the most frequently diagnosed malignant tumors in women and is also a leading cause of cancer related death. Although the prognosis for patients diagnosed at an early stage is generally favorable, the prognosis for those diagnosed at an advanced stage is generally poor. SRC-3 has long been considered a proto-oncogene in BCa and its increased expression is associated with poor prognosis in BCa patients. It acts in multiple ways on both tumor cells and the tumor microenvironment, increasing the malignant behavior of the tumor and contributing to therapy resistance. Therefore, targeting SRC-3 during BCa treatment is likely to be crucial both for inhibiting the malignant behavior of tumor cells and for overcoming therapy resistance, ultimately increasing overall treatment success. Currently, on the site “clinicaltrials.gov”, there are only two trials listed that mention SRC-3, namely NCT02311933 and NCT01327781. Both use SRC-3 as a potential clinical response marker instead of a direct target. However, it would not be surprising if new therapeutic approaches targeting SRC-3 directly for the treatment of breast cancer emerge in the near future.

## Figures and Tables

**Figure 1 cancers-15-05242-f001:**
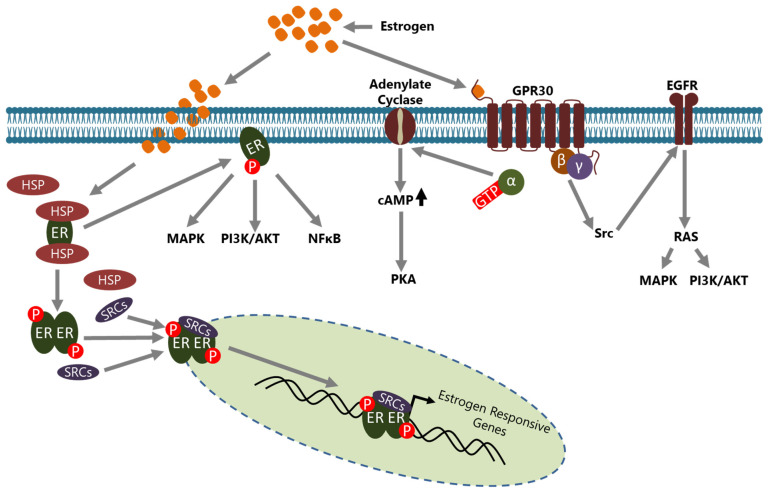
Estrogen signaling is conveyed through ERs in cells. ERs may act in a genomic or a non-genomic manner upon estrogen induction. In the genomic action mode, ERs located in the cytoplasm part from HSPs upon estrogen induction, translocate to the nucleus with some co-regulators, including the SRC family members, and consequently bind to the promoter of estrogen-responsive genes to regulate their expression directly. In the non-genomic action mode, ERs located in the cytoplasm leave HSPs upon estrogen induction and are involved in the regulation of activity of various intracellular signaling pathways. GPR30, an ER located to the cell membrane, is involved in non-genomic actions in a manner either dependent or independent from intracellular second messengers and regulates the activity of many signaling pathways. Although the non genomic action of ERs generally results in the regulation of expression of various genes, ERs do not bind directly to the promoter of those indirect target genes.

**Figure 2 cancers-15-05242-f002:**
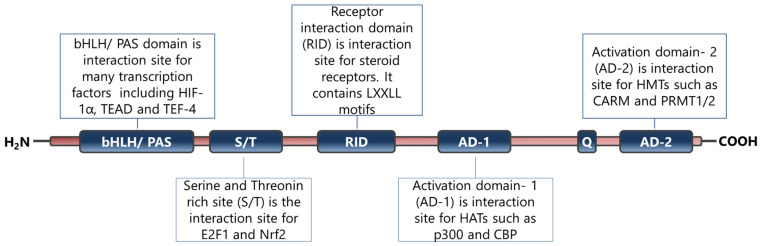
Primary structure and functional sites of SRC-3 protein. The figure has been constructed based on the data presented in refs [6,14,132].

**Figure 3 cancers-15-05242-f003:**
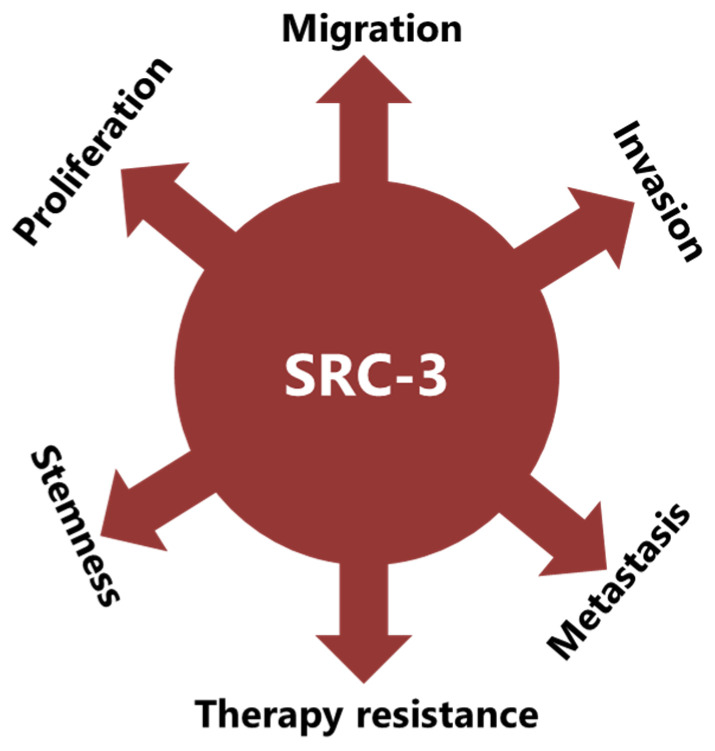
SRC-3 promotes malignancy in BCa in multiple ways. SRC-3 not only promotes the malignant behavior of tumor cells, but also changes the tumor microenvironment to an immunosuppressive phenotype which supports CSCs and contributes to therapy resistance.

**Figure 4 cancers-15-05242-f004:**
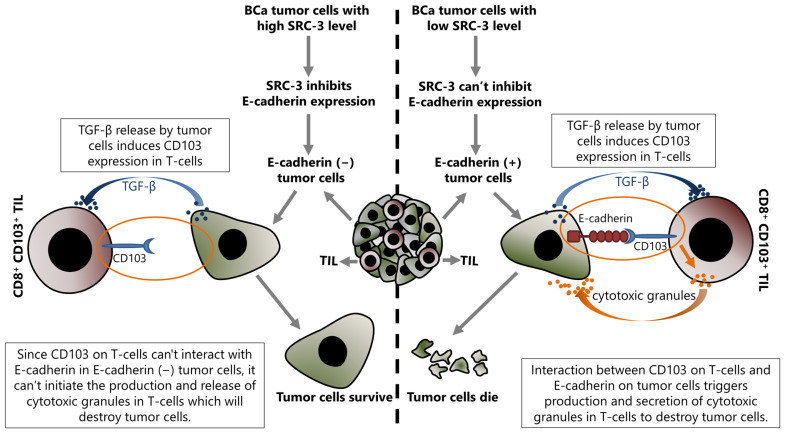
Up-regulated SRC-3 expression inhibits immunotherapy in BCa, and this effect is directly associated with the SRC-3-dependent repression of E-cadherin expression in tumor cells. Increased SRC-3 level decreases E-cadherin in tumor cells and results in abrogating the interaction between CD103 on tumor infiltrating CD8+ T-cells and E-cadherin on target (tumor) cells. Consequently, disruption of an interaction between CD103 and E-cadherin blocks the production and release of cytotoxic granules by CD8+ T-cells that would destroy tumor cells, and as a result of the lack of cytotoxic granules, the tumor cells survive.

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
