# Peer review of "Critical Roles of SRC-3 in the Development and Progression of Breast Cancer, Rendering It a Prospective Clinical Target"

_cancers, 2023, doi:10.3390/cancers15215242_

Round 1
Reviewer 1 Report
Comments and Suggestions for Authors
This is an excellent work by the authors.
General comments
The spelling and punctuation are very good. No issues were detected.
Abstract
The abstract is concise. All the necessary information about the study is included.
Background
- The information provided in the introduction is important for the comprehension of the article.
- The objective of the study is clearly mentioned.
Methods
- The methods are sufficiently explained by the authors.
Results
- The results are presented in a very extensive way.
- The figures are really helpful and necessary for the completion of the authors' work.
Discussion
- The discussion is of great quality and includes updated data.
Conclusion
From the presented data, the conclusion is complete and represents the work that the authors did.
Minor Revision
1) Despite the major advances in breast cancer surgery, there are still numerous unanswered questions regarding the histological subtype of Invasive micropapillary carcinoma.
I would like a brief discussion on Invasive micropapillary carcinoma of the breast. Please consider citing the recently published articles:
https://pubmed.ncbi.nlm.nih.gov/35310681/
and explain the roles of SRC-3 in the development and progression of this subtype, or whether there is any difference in comparison with other subtypes.
2) "HER2 is an established prognostic and predictive marker for patients with invasive breast cancer. The clinical and biological significance of HER2 overexpression in patients with ductal carcinoma in situ (DCIS) remains poorly defined. DCIS is a heterogeneous disease and some patients with DCIS will not progress to invasive breast cancer."
Add this important information and make a brief discussion on the clinical significance of HER2 expression in DCIS as well as the roles of SRC-3 in the development and progression of DCIS.
Consider citing recently published articles on this topic:
https://pubmed.ncbi.nlm.nih.gov/36352293/
Reviewer 2 Report
Comments and Suggestions for Authors
General: This study by Lokman Varisli et al systemically summarized the role of SRC-3 in development and progression of breast cancer. This review was well prepared, and give insight into potential of SRC-3 as clinical target. I have some minor concerns.
1. Although this manuscript is well prepared, the paragraph and structure need to be re-organized, for example, in the introduction section, the author introduced some background of breast cancer and estrogen signaling in the first part, while in the second part, the author also described breast cancer and estrogen; and in the 4.1 section, the author should separate stemness part as a separated section rather together with tumor microenvironment section.
2. When introducing BC tumor microenvironment, the author mainly introduce Treg cells, while other T cell subsets and their mechanism of dysfunction, should be further discussed. In addition, different subsets have different functions, role of BC tumor microenvironment in Th17 cell polarization should be discussed and summarized.
3. In the “SRC-3 affects the tumor microenvironment” part, the author cited an article indicated that RA promotes SRC3 transcription. In fact, a recent report by Wang et al indicated that SRC3 could shape MM microenvironment by inducting IL-17 expression by gamma delta T cells[1], which should be mentioned in this article and provide insight into understanding multiple role of SRC3 in shaping tumor microenvironment
4. SRC3 could promote BC by coactivating nuclear receptors as well as transcription factors, it could bind chromatin remodeling enzymes and subsequently facilitates chromatin remodeling, promoting transcriptional of malignant gene expression, this aspect should be further discussed in the review.
5. The reference of the review cited are too old, most of them are before 2010. Some latest and classic article [2] should be mentioned, and more work of BERT W. O’MALLEY should be mentioned.
Reference
[1] Wang J, Peng Z, Guo J, Wang Y, Wang S, Jiang H, Wang M, Xie Y, Li X, Hu M, Xie Y, Cheng H, Li T, Jia L, Song J, Wang Y, Hou J, Liu Z. CXCL10 Recruitment of γδ T Cells into the Hypoxic Bone Marrow Environment Leads to IL17 Expression and Multiple Myeloma Progression. Cancer Immunol Res. 2023 Oct 4;11(10):1384-1399. doi: 10.1158/2326-6066.CIR-23-0088. PMID: 37586075.
[2] Harbeck N, Penault-Llorca F, Cortes J, Gnant M, Houssami N, Poortmans P, Ruddy K, Tsang J, Cardoso F. Breast cancer. Nat Rev Dis Primers. 2019 Sep 23;5(1):66. doi: 10.1038/s41572-019-0111-2. PMID: 31548545.
Comments on the Quality of English LanguageMinor editing of English language required.
Reviewer 3 Report
Comments and Suggestions for Authors
This manuscript focuses on the critical role of Steroid Receptor Co-activator 3 (SRC-3) in breast cancer (BCa) and its potential as a therapeutic target. The review outlines SRC-3's multifaceted impact on BCa, promoting malignant behaviors in tumor cells and contributing to therapy resistance, particularly in hormone therapy and immunotherapy. In conclusion, The manuscript suggests that targeting SRC-3 holds significant promise in BCa treatment. While the manuscript provides a comprehensive overview of the role of SRC-3 in BCa and its potential as a therapeutic target, several aspects were not addressed in the manuscript:
1) The manuscript did not discuss the status of ongoing clinical trials or the existence of established therapies specifically designed to target SRC-3 in BCa.
2) It did not provide information regarding any established biomarkers or diagnostic methodologies capable of identifying BCa patients who might derive greater benefits from SRC-3-targeted therapies.
3) Given the multifaceted functions of SRC-3, the manuscript did not touch upon the potential side effects or adverse events associated with SRC-3 inhibitors or the strategies employed for SRC-3 targeting.
Reviewer 4 Report
Comments and Suggestions for Authors
In this review, the authors provide detailed information on the subject by reviewing the multifaceted effects of SRC 3 in breast malignancy and its potential as a therapeutic target. As we all know, breast cancer (BCa) is the most commonly diagnosed malignant tumour in women and is also one of the leading causes of cancer-related deaths. In this review, its stated that SRC 3 has also been shown to be overexpressed and amplified in BCa and have critical roles in the development and progression of the disease.
The authors expressed that detailed studies have shown that SRC 3 is involved in BCa pathogenesis in multiple ways, including promoting proliferation, migration, invasi on and metastasis in BCa cells. The authors have emphasised the importance of this review that SRC 3 is also involved in the creation of an immunosuppressive tumor microenvironment and promotes stemness. In regard to cancer treatment, SRC 3 is involved in the development of resistance to both endocrine therapy and immunotherapy in patients with BCa.
In conclusion on this review, the authors as a recommendation that targeting SRC 3 during BCa treatment is likely to be crucial both for inhibiting the malignant behavior of tumor cells and for overcoming therapy resistance, ultimate ly increasing overall treatment success.
In the review, detailed information is given under 7 sub-headings and with 4 well-prepared illustrations. Extensive source research has been carried out for the review. I believe that this review will provide important contributions to future studies on tumor microenvironment and cancer treatment
Author Response
Thank you very much for taking the time to review this manuscript.